# Prediction and Sound Quality Analysis of Tire Pattern Noise Based on System Identification by Utilizing an Optimal Adaptive Filter

**Sang-Kwon Lee *** , **Kanghyun An, Hye-Young Cho and Sung-Uk Hwang**

Acoustics and Vibration Signal Processing Lab., Department of Mechanical Engineering, Inha University, Incheon 22201, Korea; ankh4059@naver.com (K.A.); hycho1007@nexentire.com (H.-Y.C.); ukhwang@nexentire.com (S.-U.H.)

* Correspondence: sangkwon@inha.ac.kr; Tel.: +82-032-860-7305



**Featured Application: The proposed framework is highly suitable for designing tires with a low pattern noise.**

**Abstract:** Identifying the cause of vehicle noise is a basic requirement for the development of low-noise vehicles. The tire pattern noise depends on the tire itself and causes complex and unpredictable sounds. In pneumatic tire pattern design, the prediction technology of the tire pattern noise according to pattern shape design is important. The conventional method of predicting tire pattern noise is to simply scan the pattern shape of tire and to analyze its spectrum. However, this method has limitations because it does not consider the transfer function and precise mechanism of tire pattern noise. In this study, adaptive filter theory was applied to identify the transfer function between the grooves of patterns and measured acoustic data. To predict the waveform of an actual pattern noise in the time domain, the impulse response of this transfer function was convolved by the scanned pattern input of tires. The predicted waveform of pattern noise was validated with the waveforms of measured noise data. Finally, a sound quality index (SQI) of tire pattern noise was developed using the measured pattern noises and was applied to estimate the sound quality of pattern noise. Eventually, using the prediction method from this study, we hope to reduce the time and cost spent on tire pattern design and verification.

**Keywords:** adaptive filter; tire pattern noise; transfer function; sound quality index of pattern noise

## 1. Introduction

Noise produced by cars is important not only inside but also outside [1]. Due to the development of automotive technology and the increased interest in environmental issues, many innovative noise reduction techniques for automotive engines have been studied, especially for the development of electric cars. As the number of electric vehicles increases, the reduction of tire noise becomes more important [2,3]. Tire-road noise is the main source of noise produced by a car and is a remarkably complex phenomenon resulting from the combination of airborne and structure-borne phenomena, where the source is provided by the contact between tire and pavement [4]. In this context, it is however of paramount interest to improve the tires, but also the road pavements. Studies regarding optimal road texture and mixture design for noise abatement have been widely available in recent years [5–11]. For satisfying consumer demand on low tire noise, prediction systems for tire noise waveforms are necessary, and further elements for sound analysis and prediction steps of the process preceding the establishment of the design are also needed. The pattern noise prediction method is important because

it is difficult to change the shape and arrangement of the tire after they are determined in the beginning of the tire development process. In addition, because the pattern shape is associated with various tire performances such as groove wander, hydroplaning, and brake performance, it is an important design element. Therefore, optimal design of pattern shape is required to satisfy not only the noise performance but also other performances. Generally, the major pneumatic tire noise is divided into two types. One is the interior noise due to structure-borne path and air-borne path [12]. Airborne noise is related to compression of the air trapped within the tread of the rolling tire [13]. The other is the exterior noise caused by only the air-borne path [14]. Generally, in acoustics, the difference between structure-borne and air-borne noise is the medium of transmission, but in terms of tire noise, the difference is sometimes focused on the noise generation mechanism, which is usually misused. Strictly speaking, the latter should be called the vibro-dynamic noise and aerodynamic noise or vibro-acoustic noise and aeroacoustics noise. Pattern noise is mainly associated with the shape of the tire tread pattern, is in the range of over 500 Hz, and is a phenomenon due to air. Road noise is mainly below 500 Hz and occurs with vibration of the vehicle structure through the transfer path. Especially, pattern noise is the interior transmission sound that air delivers to the medium and is defined as air-borne noise which is over 500 Hz. A large proportion of pattern noise is correlated to air-pumping noise, which is caused by the compression and expansion of air grooves and is generated because of the grounding of the tread, the shape of the road surface, and the groove depth. Therefore, accurate noise prediction is important in the design step. A common method of estimation of the tire pattern noise is to use the blocked pitch sequence [15]. This method uses a genetic algorithm to reduce the tire air-pumping noise that is generated by the repeated compression and expansion of the air cavity between the tire pitch and the road surface. Genetic algorithms have been used to determine the optimal tire pitch sequence with low levels of tire air-pumping noise using image-based air-pumping models. However, it is difficult to predict the actual waveform with this traditional method, and in most cases, the method has limitations because it does not consider the transfer function and precise mechanism of tire pattern noise. In this study, adaptive filter theory was applied to identify the transfer function between the grooves of patterns and measured acoustic data. To predict the waveform of an actual pattern noise in time domain, the impulse response of this transfer function was convolved by the scanned pattern input of tires. The predicted waveform of pattern noise was validated with that of measured noise data. Finally, the sound quality index (SQI) of tire pattern noise was developed using the measured pattern noises and was applied to estimate the sound quality of pattern noise.

## 2. Prediction of Tire Pattern Noise Based on System Identification

### 2.1. Mechanism of Tire Pattern Noise

In general, tire pattern noise is divided into block impact noise and aerodynamic air-pumping noise. Impact noise is generated by the impact of the road surface with the tire pattern block. It is especially generated because of air-pumping when the pattern groove is in contact with the ground. Air-pumping was first proposed in 1971 by Hayden [16], and its generation mechanism is represented in Figure 1. Air pumping is the noise generated because of the compression and subsequent expansion of the air trapped between the tire tread and the road surface [4]. Air in the grooves is released rapidly to the outside because of compression of the groove, and at the moment the groove and road surface separate, air flows back into the groove because of the increase in its volume. This phenomenon is an important factor in pattern noise associated with the groove factor.

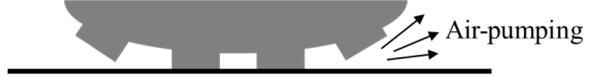

**Figure 1.** Air-pumping of tread grooves.

### 2.2. System Identification Based on Adaptive Filter Theory

Simple data scanning for estimating pattern noise, which is the conventional method, predicts the pattern noise only using pattern shape and has limitations because it does not consider the transfer function and precise mechanism. To improve this method, image processing was applied to predict tire pattern noise by considering the effect of groove depth on noise. In this study, the adaptive filter theory was also applied to calculate the transfer function between the pattern groove image data and the measured data. Figure 2 shows the procedure used for the application of the adaptive filter to predict actual pattern noise in the time domain. First, the images of the tire pattern were processed to predict the input force exciting the tire. Second, an optimal adaptive filter was designed to identify the transfer function between the input force and noise output. To design the optimal filter, Wiener's adaptive filter was applied [17]. The output data of the optimal filter, *f*, should be the same as the measured noise to minimize the error. The transfer function of the optimal filter *f* becomes that of the transfer path between the excitation force and the measured noise, as shown in Figure 2. In this study, the tire pattern noise was measured in a semi-anechoic chamber, as shown in Figure 2. A test tire was installed on a roller which was controlled by a dynamometer. Pattern noise was measured at 1 m from the center of the tire. Three positions were selected for this test. All data presented in this study were measured at 1 m from the tire center in the front of the tire. Four test tires were used, and their input forces were scanned using image processing.

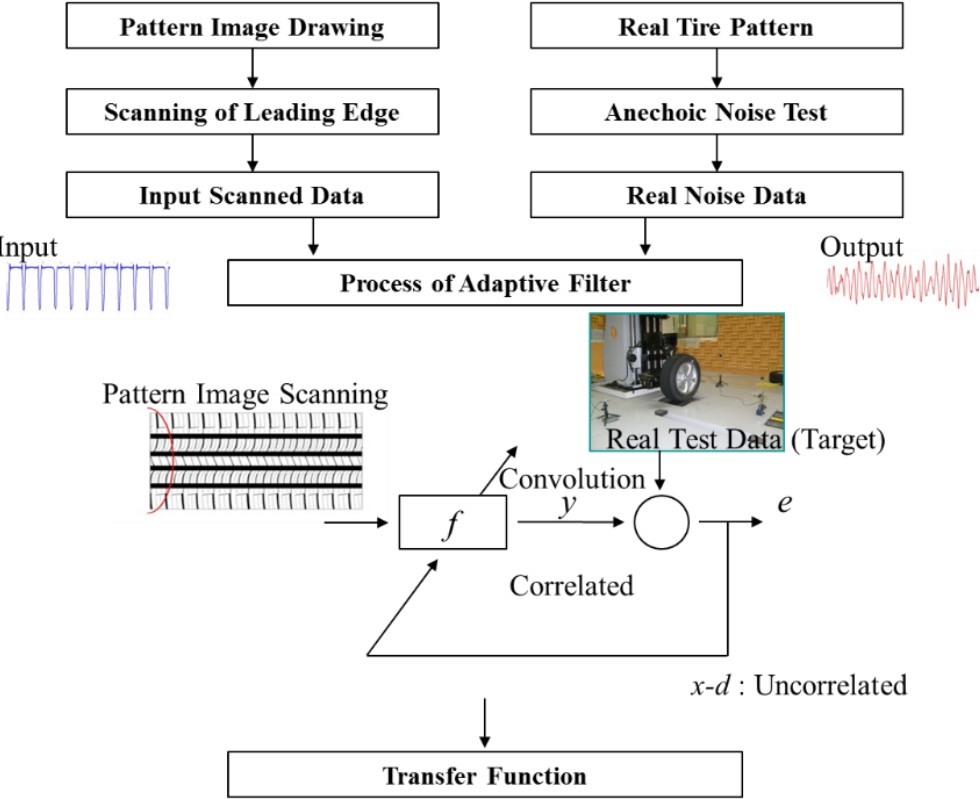

**Figure 2.** Prediction of transfer function based on system identification using an optimal adaptive filter.

To predict the noise radiated from the new tire pattern, the transfer function or impulse response function of the optimal filter waveform can be used. The noise from the new tire pattern is predicted by convoluting the excitation force obtained throughout the image processing for the new tire pattern and the impulse response of the optimal filter.

## 2.3. Optimal Adaptive Filter Theory

The adaptive filter was developed considering the "adaptive linear combiner" shown in Figure 3.

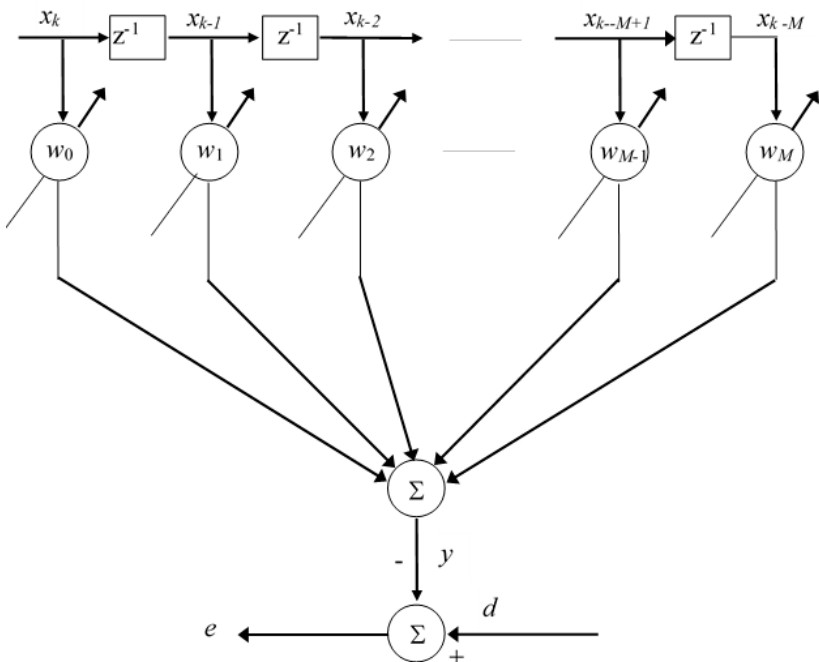

**Figure 3.** Linear combiner tap delayed adaptive filter.

The adaptive filter output at discrete time $k$ is defined by the linear convolution sum,

$$y = \mathbf{x}^T \mathbf{w} = \mathbf{w}^T \mathbf{x} \tag{1}$$

where $\mathbf{x}$ = input force vector; $\mathbf{w}$ = tap weight vector of adaptive filter $f$; $y$ = predicted noise which is output sequence of filter.

To optimize the filter, we chose to minimize the mean square value of the estimate error $e$, which is defined as follows

$$e = d - y \tag{2}$$

where $d$ is the desired response or target data. We may define the *cost function* as the mean squared error $(\xi)$, which is evaluated by taking the expectation of the square of the error (e).

$$\xi = E\left[\left(e^2\right)\right] = E\left[\left(d^2\right)\right] + \mathbf{w}^T \mathbf{R}\mathbf{w} - 2\mathbf{p}\mathbf{w} \tag{3}$$

where $\mathbf{R} = E\left[\mathbf{x}\mathbf{x}^T\right]$ and $\mathbf{P} = E[d\mathbf{x}]$. $\mathbf{R}$ is the auto correlation matric of input vector, and $\mathbf{P}$ is the cross-correlation vector between the input vector and the desired value. This equation is a quadratic equation in terms of the weight vector ($\mathbf{w}$). Figure 4 shows a typical performance surface for a filter of length ($M = 2$). The gradient of the mean square error, $\nabla(\xi)$, can be obtained by differentiating Equation (3) with respect to the weight vector $\mathbf{w}$ to obtain the column vector

$$\nabla(\xi) = \frac{\partial(\xi)}{\partial \mathbf{w}} = -2e\,\mathbf{x} = 2\mathbf{R}\mathbf{w} - 2\mathbf{p} \tag{4}$$

The *cost function* $(\xi_k)$ attains its minimum value when the gradient of this quadratic equation is zero:

$$\mathbf{W}^o = \mathbf{R}^{-1}\mathbf{p}. \tag{5}$$

This is the "Wiener" optimal filter. The optimal tap-weighting vector of the adaptive filter $f$ is shown in Figure 2.

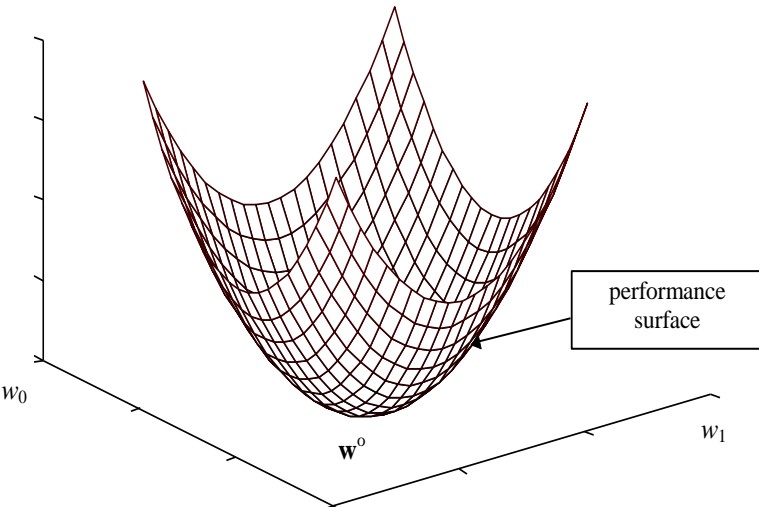

**Figure 4.** Performance surface of a *cost function*.

### 2.4. Estimation of Pattern Noise Using the Optimal Adaptive Filter

We applied the developed prediction method on the prediction of tire pattern noises of four commercial tires and verified it with measured pattern noise data. The optimal filter for the prediction model was obtained by applying Equation (5) to the measured data. The filter output is the predicted pattern noise. The comparison between the predicted noises and the measured pattern noises is shown in Figure 5. There are little differences between the measured data and the estimated data. In Figure 5, the predicted noise is represented by a solid line and the measured value is represented by a dotted line. The power spectral density (PSD) [18] for those noises was calculated, and its plot is shown in Figure 6. Figure 6 shows the frequency characteristics for each pattern noise. According to previous studies [15], because the tire pattern noise occurs in the frequency band of 500–2000 Hz, the magnitude of tire pattern noise in this frequency band is represented. The predicted noise is represented by a solid blue line, and the measured value is represented by a dotted red line. According to these results, there is a slight difference between the sound waves of the measured pattern noise and those of the predicted pattern noise. However, waveform and spectrum characteristics of both pattern noises are similar. Even if there is a slight difference, in the early design stage, to estimate the sound quality of pattern noise, we need an estimation model of pattern noise associated with pattern shape. The model was validated using 10 tires which were not used for model development to check if the model can be used to estimate sound quality of pattern noise, as presented in Section 3. Our results show that although it does not provide an exact solution as an in-situation application method, it can be concluded that the proposed method can predict the tire pattern noise in the early design stage without conducting experiments. In addition, the predicted sound waves can be used for sound quality analysis of pattern noise at the early design stage using the sound quality index (SQI). The SQI of pattern noise was developed using the measured tire sounds. The index can predict the sound quality of pattern noise. It was developed based on subject evaluation and sound metrics of the measured pattern noises. The sound metrics of pattern noise should correlate with subjective evaluation results. A detailed SQI development process for pattern noise will be presented in the next section.

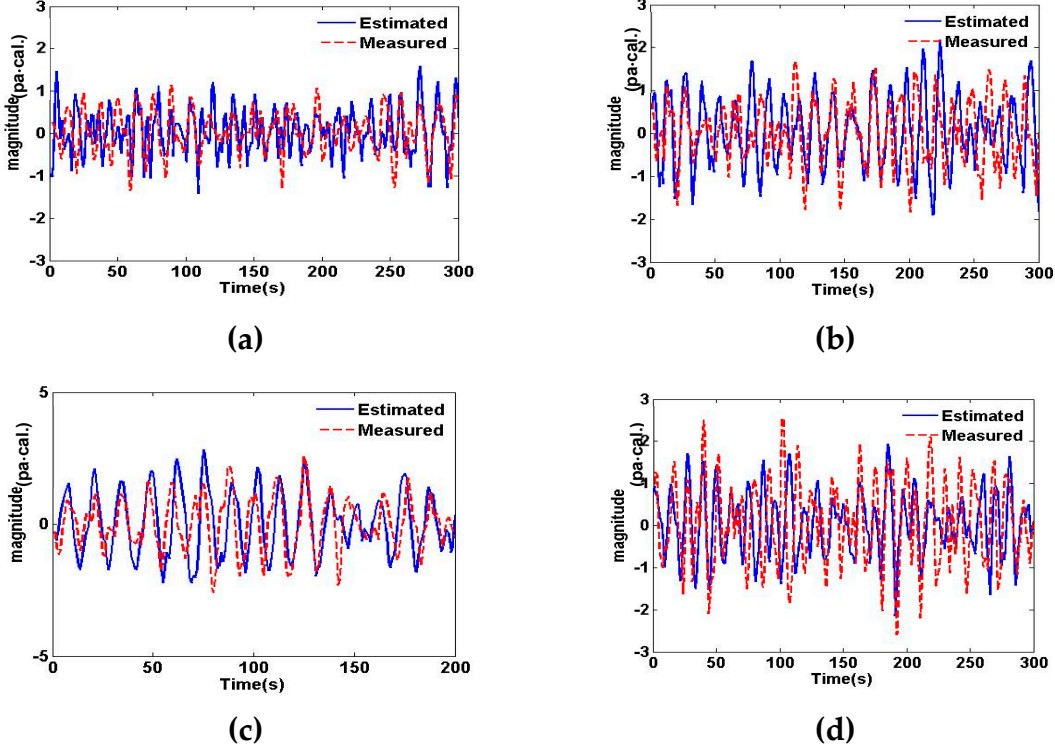

**Figure 5.** Time data comparison of estimated signal between measured signals. (**a**) Pattern A, (**b**) Pattern B, (**c**) Pattern C, and (**d**) Pattern D; (y axis unit: Pascal, *calibration factor* = $1.58 \times 10^3$).

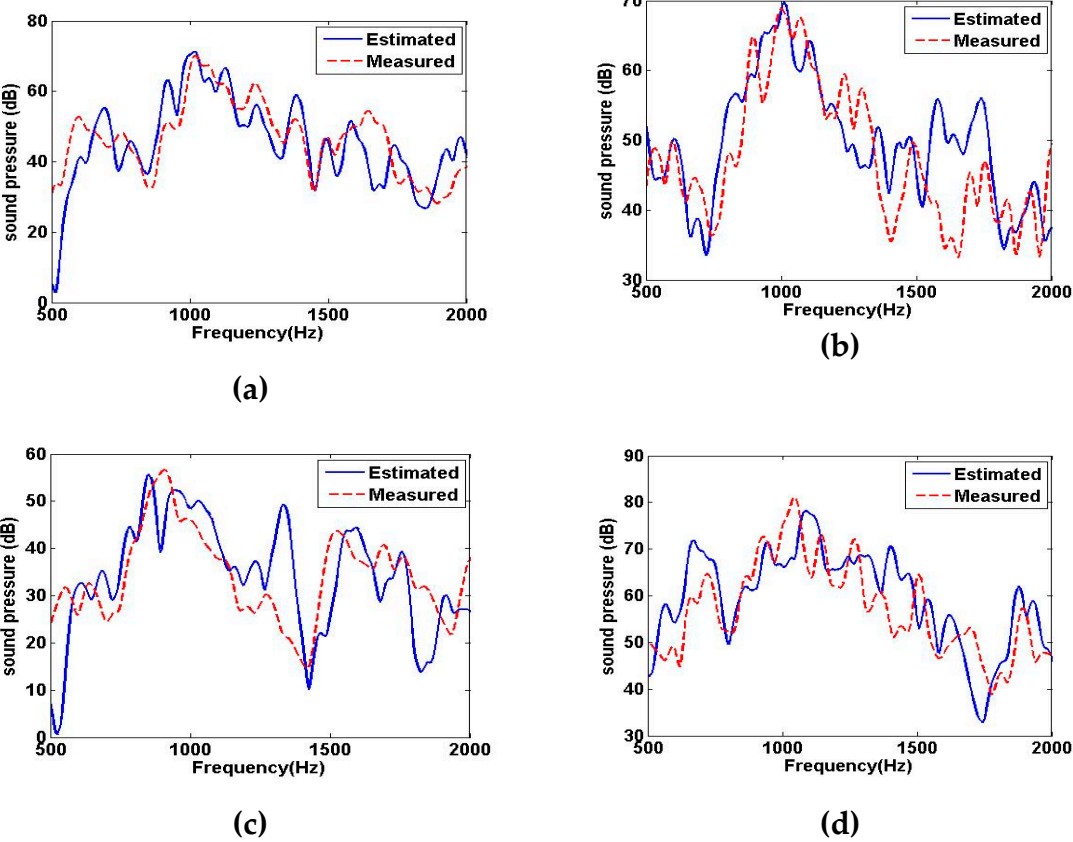

**Figure 6.** Power spectral density (PSD) comparison of estimated signal between measured signals. (**a**) Pattern A, (**b**) Pattern B, (**c**) Pattern C, and (**d**) Pattern D; (y axis dB_*reference* = $2 \times 10^{-5}$ Pascal).

## 3. Sound Quality Analysis of Pattern Noise

### 3.1. Jury Evaluation

To evaluate the sound quality of the pattern noise of a new tire pattern, the SQI is needed as the objective method. To develop the SQI of tire pattern noise, subjective evaluation was performed for 10 commercial tire patterns. Pattern noises of commercial tires were measured at the semi-anechoic chamber, as shown in Figure 7.

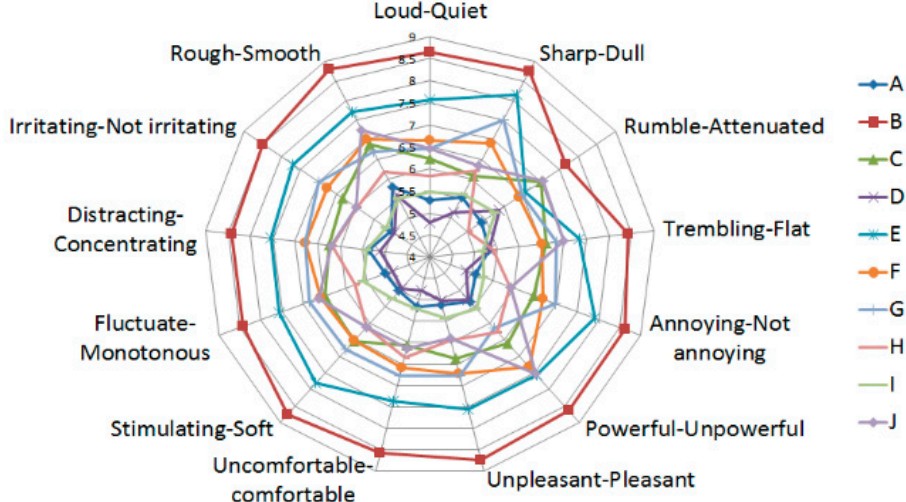

**Figure 7.** Results of subjective evaluation.

The noises were measured at 1 m from tire center in the front. In the subjective evaluation, a questionnaire was used. It is important to find query pairs that can comprehensively express tire noise. With the reference to the results of various studies [19], 13 query pairs related to tire pattern noise were used. The jury test for the selected 13 query pairs was carried out by the semantic differential method (SDM). Twenty students participated for the subjective evaluation. To improve the reliability and accuracy of jury evaluation, a prior training, which takes approximately 30 min, was conducted. In SDM, 4–9 points for rating were applied. Figure 7 shows the results of the subjective evaluation for the tire pattern noise in a radial graph to present the subjective score of each pattern in each query pair. One query pair was the major factor that represented the attributes for sound quality of the pattern noise. The 'pleasant-unpleasant' attribute, which is obtained by principal analysis, was selected [20].

### 3.2. SQI of the Tire Pattern Noise

To evaluate the sound quality of pattern noise objectively, a prediction model was developed using psychoacoustic metrics [21]. There are several metrics in this field [22]. The psychoacoustic metrics for 10 pattern noises and their correlation with subjective evaluation was calculated. It was found that loudness and sharpness were the major metrics affecting the subjective evaluation of tire pattern noise. Figure 8 shows the correlation between psychoacoustic metrics and subjective rating. Finally, to develop the predictive model, the multiple regression method [23] was employed. The results of the multiple regression are listed in Table 1.

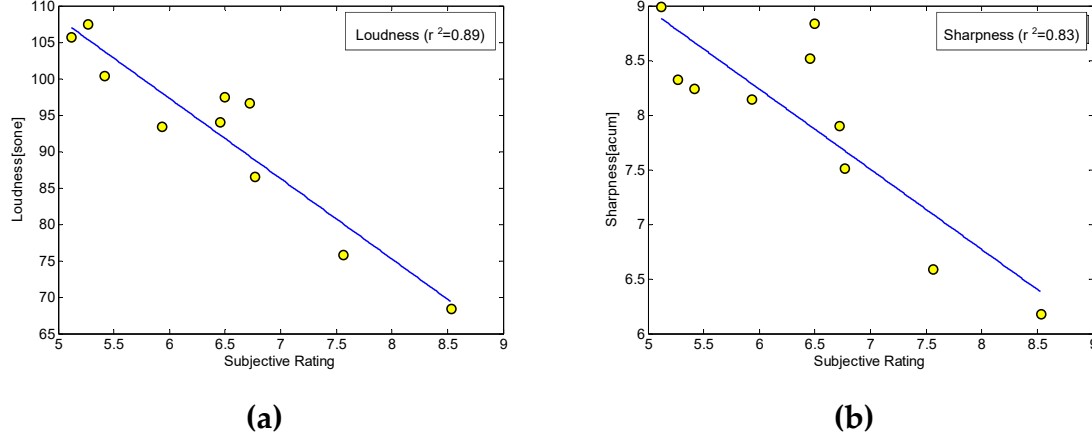

**Figure 8.** Correlation plot of the regression analysis and sound metrics; (**a**) loudness vs. subjective evaluation; (**b**) sharpness vs. subjective evaluation.

**Table 1.** Results of multiple linear regression analysis.

| Model | Non-Standardization Coefficient | | Standardization Coefficient | $t$ | $p$ |
| --- | --- | --- | --- | --- | --- |
| | $B$ | Standard Error | $\beta$ | | |
| (constant) | 13.743 | 1.151 | | 11.943 | 0.000 |
| Loudness | −0.088 | 0.026 | −1.027 | −3.328 | 0.013 |
| Sharpness | −0.104 | 0.358 | 0.090 | 0.292 | 0.0779 |

In Table 1, the non-standardized coefficient, *B*, is the coefficient of the SQI. The standard error denotes the value of standard deviation divided by the square root of the number of samples. The standardized coefficient, *β*, is the influence of the SQI. If the absolute value of *β* is high, it has a high impact on the results. In contrast, if the absolute value of *β* is low, then the influence of each model is low. The *t*-value is given by *B*/standard-error. It gives the location of the *t*-value on the distribution curve, so probability can be estimated by the *t*-value. From the *B* value in Table 1, the SQI for the tire pattern noise from multiple linear regression analysis can be given as follows:

$$SQI = 13.743 - 0.088{\cdot}L - 0.104{\cdot}S \tag{6}$$

where *L* is loudness and *S* is sharpness. In Table 1, the *p*-value for sharpness is 0.779 and is larger than 0.05. The reason of this high *p*-value is the high mutual correlation of loudness and sharpness (correlation coefficient: 0.94). This high *p*-value indicates that sharpness does not add information to the model and sharpness is excluded. Therefore, Equation (6) is replaced by

$$SQI = 162.89 - 10.97{\cdot}L \tag{7}$$

*3.3. Sound Quality Analysis Based on Predicted Pattern Noises*

The sound metrics (loudness and sharpness in this study) were compared with the simple scan method [15] to analyze the effect of the proposed method. The coefficient of determination $r^2$ of the simple scan method and adaptive filter method are compared, as shown in Figures 9 and 10. In the figures, the y-axis represents estimated data, the x-axis is the measured data, and the yellow dots represent the value of each pattern. Figure 7 shows data from the simple data scan method, and Figure 8 shows data from the adaptive filter method proposed in this study. In the plot, the correlation value of the adaptive filter is significantly higher than that of the simple scan method. This means that the adaptive filter method is more effective for estimating tire pattern noise.

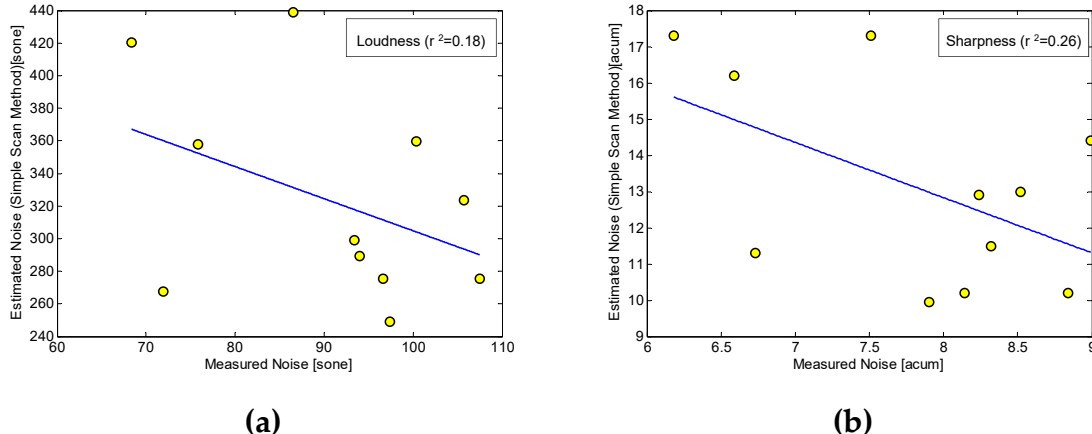

**Figure 9.** Estimating ability of the simple scan method; (**a**) estimation of loudness; (**b**) estimation of sharpness.

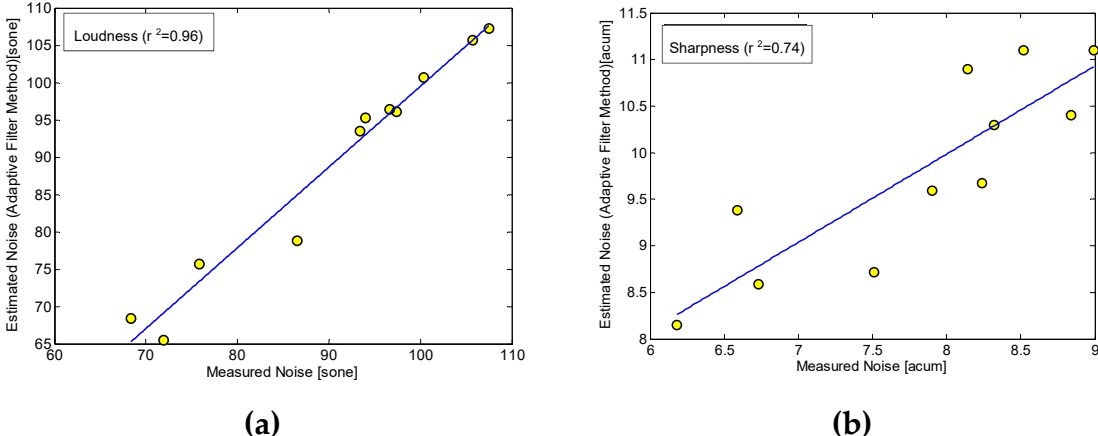

**Figure 10.** Estimating ability of the adaptive filter method; (**a**) estimation of loudness; (**b**) estimation of sharpness.

In Figure 9, the loudness and sharpness values of the simple scan data method and their correlation values are presented. The estimating abilities of the simple scan method for loudness and sharpness are 0.18 and 0.26, respectively. In Figure 10, the loudness and sharpness values of the adaptive filter method and their correlation values are presented. The estimating abilities of the adaptive filter method for loudness and sharpness are 0.96 and 0.74, respectively. From these results, loudness and sharpness are highly accurately estimated. There are other data for sound metrics, but the most influential factor, loudness, has the highest correlation coefficient. As a result, loudness and sharpness are applicable factors for the SQI of tire pattern noise.

## 4. Discussion

Our estimating system using convolution and adaptive filter theory proved the efficiency of this adaptive filter method on commercial tire patterns. In the case of comparison for sound metrics, the ability of the proposed estimating system is 98% for loudness and 86% for sharpness. As a result, data estimated for loudness and sharpness are applicable for the SQI of tire pattern noise. In this study, we developed a tire pattern noise estimating system and calculated the SQI of tire pattern noise. From the adaptive filter method, we can obtain the waveform data of each tire pattern before manufacturing the tire. Especially, the most influential factor, loudness, had the highest correlation coefficient. From the subjective evaluation, if overall sound pressure level is low, people had a tendency to like the noise.

This sound pressure level is related to the groove depth of a tire, so we can prevent poor sound tire patterns in advance.

## 5. Conclusions

To estimate tire pattern noise signals more accurately, convolution and adaptive filter theory was applied, and the results were verified by comparing the measured data with the estimated data from tire pattern images. We also compared the proposed method with the traditional method (simple scan method), and the adaptive filter method was found to more efficiently predict the tire pattern noise. By using this method, we can obtain more precise wave file data from tire pattern images and its sound quality. Although the estimated sound pressure does not exactly correspond to the measured sound pressure, the loudness of both pressures correspond, and the value of $r^2$ is 96. It can be concluded that the proposed method can be used for prediction of sound quality of pattern noise in the early design stage. For better estimation of sound pressure based on adaptive filters, deep learning with more data can be applied. Eventually, using our proposed prediction method, the time and cost spent on development of tire pattern shapes can be effectively reduced and the quality of tires can be improved.

**Author Contributions:** S.-K.L. contributed to simulation and algorithm development and writing the paper. K.A. contributed to the vehicle experimental test, H.-Y.C. and S.-U.H. supplied test tires and anechoic chamber for tire noise test.

**Funding:** This work was supported by Mid-career Researcher Program through NRF of Korea grant funded by the MEST (No. 2019R1A2B5B02069400).

**Acknowledgments:** This work was supported by Inha University.

**Conflicts of Interest:** The authors declare no conflict of interest.

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
