# Peer review of "Prediction and Sound Quality Analysis of Tire Pattern Noise Based on System Identification by Utilizing an Optimal Adaptive Filter"

_applsci, doi:10.3390/app9193995_

Round 1

Reviewer 1 Report

The paper deals vehicle noise using an interesting approach based on adaptive filtering techniques.

Several issues arise when reading the paper.

·         Firstly, English language requires a deep revision, since several mistakes are present. A non-comprehensive list is hereby provided:

1)      Generally, the preposition for requires the ing form of verbs (e.g. line 31, for satisfy need to be replaced by “for satisfying”, line 73 “for estimate” -> “for estimating”);

2)      Several sentences are written in an unclear way and their meaning is difficult to grasp, such as line 35 “as well as noise but groove wander, hydroplaning, snow, brake performance is associated with such a variety of different tire performance is important design elements”, line 191 “the estimating effect analysis…”

3)      Minor errors could be a general spelling check, which should be coherent among the different English standards: if the American spelling of “tire” is adopted, then this convention should be adopted throughout the text and therefore “optimization” should be preferred over the British “optimisation”. Line 137, the “predicated” sound waves should be “predicted”.

4)      Several periods should be revised since they are missing the main sentence (line 130 “Where tire pattern…, so it is considered…”) or the verb (line 64 “which shown in figure 1”, line 187 “which significantly high”).

Improving the use of English language would increase the intelligibility of the paper.

·         “Structure-borne” and “airborne” are mentioned in this paper, which might be inaccurate. These two terms are accurate and meaningful for vehicle interior noise depending on the transfer path. For vehicle exterior noise (CPX), all the noise should be called “airborne”. Also refer to the paper below on Page2429.

https://doi.org/10.21595/jve.2018.19935

“Generally, in acoustics the difference between structure-borne and airborne noise is in the medium of transmission, but in terms of tire noise, the difference is sometimes focused on the noise generation mechanism, which is kind of misused. Strictly speaking, the latter should be called as the vibro-dynamic noise and aerodynamic noise, or vibro-acoustic noise and aeroacoustic noise”.

·         A section concerning “state of the art” is also missing, despite several tyre models have been developed. In particular, a description of the “simple scan method” should be presented in the paper, since it is used for comparisons.

·         It is also incorrect to state that “Air pumping is the moment of contact”. Air pumping is a noise generation mechanism, due to the compression and subsequent expansion of the air trapped between the tyre tread and the road surface. See also Sandberg, U., & Ejsmont, J. (2002). Tyre/road noise. Reference book.

·         The “comparison plots” (Figure 5) are not enough to determine the validity of the model. Moreover, it is unclear if the data shown has been used for modelling and validation or two separate datasets were used, as it should be. The PSD comparison (Figure 6) also shows a general agreement between estimated and measured values, however plot (b) shows a plateau for estimated values at 1500-1800 Hz and plot (c) shows a rather sharp peak at 1250. These differences should be addressed in the text.

·         The modelling section needs to be revised. The multivariate regression in table 1 shows the wrong coefficient for S (0.104), since the plot shows clearly that S is negatively correlated with the subjective rating. When analyzing the goodness-of-fit of a multivariate regression, it is important to analyze not only the global p-value of the fit, but of single variables too. The p-value for the sharpness is 0.779: such a high p-value indicates that its presence does not add information to the model and this variable should be excluded. It is arguable that Sharpness and Loudness probably are already mutually correlated: please show a plot of these two variables. It is also noticeable that usually r^2 values are presented using the decimal dot, rather than their % form.

·         The discussion on p-value can be deleted from text, since it is rather confusing.

·         Figures 8 and 9 show separate plots for estimated noise from loudness and sharpness: a better way of showing the validity of the model could be the equality line plot, not a separate plot for loudness and sharpness. The interpretation of the correlation coefficient as an “estimating ability” is also misleading and it should be replaced by more appropriate quantities such as the coefficient of determination r^2, which suggests the explained variance.

·         Discussions must be revised and extended and the same applies to the conclusions, which lack further developments and a presentation of the model limits and its strong points.

Author Response

Thank you for your kind comments for the improvement of quality of paper. Authors revised the paper as reviewer recommends and marks the revised points with yellow color

The paper deals vehicle noise using an interesting approach based on adaptive filtering techniques.

Several issues arise when reading the paper.

Firstly, English language requires a deep revision, since several mistakes are present. A non-comprehensive list is hereby provided:

Response: Native speaker checked and revised some sentence and word rammer.

1)      Generally, the preposition for requires the ing form of verbs (e.g. line 31, for satisfy need to be replaced by “for satisfying”, line 73 “for estimate” -> “for estimating”);

Response: The words are corrected in the text (line 33, line 88).

2)      Several sentences are written in an unclear way and their meaning is difficult to grasp, such as line 35 “as well as noise but groove wander, hydroplaning, snow, brake performance is associated with such a variety of different tire performance is important design elements”, line 191 “the estimating effect analysis…”

Response: The sentences are revised in the test (line 38).

3)      Minor errors could be a general spelling check, which should be coherent among the different English standards: if the American spelling of “tire” is adopted, then this convention should be adopted throughout the text and therefore “optimization” should be preferred over the British “optimisation”. Line 137, the “predicated” sound waves should be “predicted”.

Response: The words are corrected in the text.

4)      Several periods should be revised since they are missing the main sentence (line 130 “Where tire pattern…, so it is considered…”) or the verb (line 64 “which shown in figure 1”, line 187 “which significantly high”).

Response: The sentences are revised in the test (at line 141, 77 and 138).

Improving the use of English language would increase the intelligibility of the paper.

“Structure-borne” and “airborne” are mentioned in this paper, which might be inaccurate. These two terms are accurate and meaningful for vehicle interior noise depending on the transfer path. For vehicle exterior noise, all the noise should be called “airborne”. Also refer to the paper below on Page2429. https://doi.org/10.21595/jve.2018.19935. “Generally, in acoustics the difference between structure-borne and airborne noise is in the medium of transmission, but in terms of tire noise, the difference is sometimes focused on the noise generation mechanism, which is kind of misused. Strictly speaking, the latter should be called as the vibro-dynamic noise and aerodynamic noise, or vibro-acoustic noise and aeroacoustic noise”.

Response: The sentence is revised (line 43) and the recommended paper is cited in reference [5].

A section concerning “state of the art” is also missing, despite several tyre models have been developed. In particular, a description of the “simple scan method” should be presented in the paper, since it is used for comparisons.

Response: The scan method is explained in line 56 with additional sentence.

It is also incorrect to state that “Air pumping is the moment of contact”. Air pumping is a noise generation mechanism, due to the compression and subsequent expansion of the air trapped between the tyre tread and the road surface. See also Sandberg, U., & Ejsmont, J. (2002). Tyre/road noise. Reference book.

Response: The sentences are revised (line 77) and the recommended reference is cited in reference [8].

The “comparison plots” (Figure 5) are not enough to determine the validity of the model. Moreover, it is unclear if the data shown has been used for modelling and validation or two separate datasets were used, as it should be. The PSD comparison (Figure 6) also shows a general agreement between estimated and measured values, however plot (b) shows a plateau for estimated values at 1500-1800 Hz and plot (c) shows a rather sharp peak at 1250. These differences should be addressed in the text.

Response: The optimal filter for the prediction model was obtained by applying the Eq. 5 to the measured data. The filter output is the predicted pattern noise. As explained in the line 153, the model is not exact solution. However, in early design stage, in order to estimate the sound quality of pattern noise, we need an in-situation estimation model of pattern noise associated pattern shape. The validation of model, if we can use it for the estimation of sound quality of pattern noise, was presented in section 3 using 10 tires which were not used for model development.  

The modelling section needs to be revised. The multivariate regression in table 1 shows the wrong coefficient for S (0.104), since the plot shows clearly that S is negatively correlated with the subjective rating. When analyzing the goodness-of-fit of a multivariate regression, it is important to analyze not only the global p-value of the fit, but of single variables too. The p-value for the sharpness is 0.779: such a high p-value indicates that its presence does not add information to the model and this variable should be excluded. It is arguable that Sharpness and Loudness probably are already mutually correlated: please show a plot of these two variables. It is also noticeable that usually r^2 values are presented using the decimal dot, rather than their % form.

Response: The coefficient for S in table 1 is corrected. The sharpness is excluded since the p-value for the sharpness is high and the mutual correlation of loudness and sharpness is presented at line 206 in text. The correlation coefficient is replaced by r^2 in the plot and text.

The discussion on p-value can be deleted from text, since it is rather confusing.

Response: As reviewer recommends, the discussion on p-value is deleted in text.

Figures 8 and 9 show separate plots for estimated noise from loudness and sharpness: a better way of showing the validity of the model could be the equality line plot, not a separate plot for loudness and sharpness. The interpretation of the correlation coefficient as an “estimating ability” is also misleading and it should be replaced by more appropriate quantities such as the coefficient of determination r^2, which suggests the explained variance.

Response: The correlation coefficient is replaced by r^2 in the plot and text. ·         Figures 8 and 9 cannot be represented by the equality line plot since the scale of loudness is different with that of sharpness. “estimating ability” is replaced by coefficient of determination.

Discussions must be revised and extended and the same applies to the conclusions, which lack further developments and a presentation of the model limits and its strong points.

Response: Discussions must be revised and extended in the text ( line 249)

Reviewer 2 Report

applsci-586613

This reviewer believes that the paper presented deserves to be published in Applied Sciences  : the introduction contains sufficient references, the research approach is adequate, the method is well presented and the conclusions are supported by data.

I just want to point out some minor revisions that must be resolved:

1. In figure 5, units on the "y" axis
2. In figure 6, what is the reference for dBs of the y-axis?

Author Response

Thank you for your kind comments for the improvement of quality of paper. Authors revised the paper as reviewer recommends and marks the revised points with yellow color

This reviewer believes that the paper presented deserves to be published in Applied Sciences: the introduction contains sufficient references, the research approach is adequate, the method is well presented, and the conclusions are supported by data.

I just want to point out some minor revisions that must be resolved:

In figure 5, units on the "y" axis

Response: As recommended it is presented

In figure 6, what is the reference for dBs of the y-axis?

Response: As recommended it is presented

Round 2

Reviewer 1 Report

The paper generally presents a great improvement compared to the first manuscript. However, reading the paper again and in a better shape, made me find a new issue:
I believe the paper reports a very low number of references and this should be fixed. In the specific, in the introduction is important to specify and give space to the environmental noise topic. Noise produced by car is important not only inside (Zhang, L., Kang, J., Luo, H., & Zhong, B. (2018). Drivers’ physiological response and emotional evaluation in the noisy environment of the control cabin of a shield tunneling machine. Applied Acoustics, 138, 1-8), but also outside.
Thus, a period like the following can be added for improve the quality:
“Tyre-road noise is the main source of noise produced by a car and is a remarkably complex phenomenon resulting from the combination of airborne and structure-borne phenomena, where the source is provided by the contact between tyre and pavement (Sandberg U, Ejsmont J (2002) Tyre/road noise reference book, INFORMEX, Kisa, Sweden.). Airborne noise is related to compression of the air trapped within the tread of the rolling tyre (Morgan, P. A., Phillips, S. M.; Watts, G. R. The localisation, quantification and propagation of noise from a rolling tyre. TRL Limited, 2007.). In this context, it is however of paramount interest to improve the tyres, but also the road pavements. Studies regarding optimal road texture and mixture design for noise abatement are widely available in recent years (Losa, M., Leandri, P., & Licitra, G. (2013). Mixture design optimization of low-noise pavements. Transportation Research Record, 2372(1), 25-33; Hamet, J. F., & Klein, P. (2000, August). Road texture and tire noise. In Proc. inter noise (pp. 178-183); Losa, Leandri, Bacci, 2010; Licitra, G., Cerchiai, M., Teti, L., Ascari, E., Bianco, F., & Chetoni, M. (2015). Performance assessment of low-noise road surfaces in the Leopoldo project: comparison and validation of different measurement methods. Coatings, 5(1), 3-25.; Praticò, F. G. (2014). On the dependence of acoustic performance on pavement characteristics. Transportation Research Part D: Transport and Environment, 29, 79,87; Licitra, Gaetano, et al. "The influence of tyres on the use of the CPX method for evaluating the effectiveness of a noise mitigation action based on low-noise road surfaces." Transportation Research Part D: Transport and Environment 55 (2017): 217-226.; Licitra, G., Moro, A., Teti, L., Del Pizzo, A., & Bianco, F. (2019). Modelling of acoustic ageing of rubberized pavements. Applied Acoustics, 146, 237-245.)

Author Response

Thank you for your kind review. We attached the response as a file.
